# BINARY HYPERBOLIC EMBEDDING

## ABSTRACT

As datasets continue to grow, vector-based search becomes more storage and compute intensive, requiring large-scale systems to support retrieval. Proposed solutions range from quantization techniques that balance speed and accuracy, to hashing methods that learn compact binary representations. This paper promotes the use of hyperbolic space for its compact nature whilst overcoming its slow retrieval via binarization. Specifically, we address hyperbolic space's inherent slowness by proving that its complex similarity calculations can be equated to a binary XOR operation. Our approach allows for 90% less storage and at least 4.7 times faster search while maintaining performance of full-precision Euclidean embeddings.

## 1 INTRODUCTION

Compressed representations benefit information retrieval, as they greatly reduce storage requirements for data embeddings. Therefore, this property is desirable in many practical scenarios, where retrieval-by-embedding needs to be fast or performed on large collections. Prior work has shown that considerable speed-ups can be obtained for Euclidean representations by binarizing (Cai et al., 2020; Jacob et al., 2018; Kim et al., 2021), or by hashing (Wang et al., 2018; Shen et al., 2020; Hoe et al., 2021) representations on top of a learned network. These approaches do so by splitting the Euclidean representations into regular grids. In contrast, hyperbolic representations naturally allow for lower-dimensional representations (Long et al., 2020) due to their compact nature. Unfortunately, this compactness comes with a trade-off: reduced computation speed due to complex similarity calculations (Peng et al., 2021). In this work, we show that this trade-off can be overcome through binarization, thereby unlocking the full potential of hyperbolic embeddings.

Hyperbolic deep learning has quickly gained traction in the field. Primarily, because it allows embedding hierarchies with minimal distortion (Nickel & Kiela, 2017), vastly outperforming Euclidean hierarchical embeddings (Ganea et al., 2018b; Sala et al., 2018). These benefits have been shown for various research problems, from graph networks (Chami et al., 2019; Dai et al., 2021; Liu et al., 2019) to reinforcement learning (Cetin et al., 2023). Specifically, for image and video representation learning, where search is often performed, hyperbolic geometry allows for fewer embedding dimensionalities (Liu et al., 2020; Ermolov et al., 2022) and better hierarchical learning (Nickel & Kiela, 2017; Ganea et al., 2018b; Sonthalia & Gilbert, 2020). Despite these advantages, hyperbolic embeddings have not been a viable option for retrieval-by-embedding, as calculating the distance between embeddings involves multiple slow vector operations.

This paper introduces binary hyperbolic embedding, a binarization approach that addresses the core limitation of hyperbolic embeddings for retrieval. As contributions, we first prove how to approximately binarize distances in the Poincaré model of hyperbolic space at high compression rates. Second, we demonstrate how to use the binarized distance in a hyperbolic hierarchical embedding network to get both fast binarized search and compact hyperbolic embeddings.

Our contributions are as follows:

- We prove that slow hyperbolic distance computation is equivalent to fast Hamming distance computation with propose binary encoding.

- With our binary hyperbolic embedding, we are able to induce a large speed-up with minimal loss in performance, thus obtaining at least $4.7 \times$ faster speeds than full-precision euclidean embeddings.

- We further show that these benefits hold across a variety of settings, including the ability to incorporate hierarchical knowledge.

Our work makes it possible to perform fast search in binarized hyperbolic space, making hyperbolic embeddings a viable alternative for large-scale search and retrieval.

## 2 RELATED WORK

**Learning with binarized and quantized embeddings.** Compressing representations is a common task in retrieval and search. For a query, the goal is to find the nearest neighbors in a collection. Since search typically needs to occur on-the-fly (Yuan et al., 2020; Wang et al., 2018) or on huge collections (Jang & Cho, 2021; Chen et al., 2023), it is imperative to efficiently embed queries and data collections. The efficiency of an embedding can be expressed in bits, where fewer bits can ultimately only be obtained in two ways: using fewer embedding dimensions (Cao et al., 2020; Hausler et al., 2021) and/or using fewer bits per dimension (Choukroun et al., 2019; Yao et al., 2022; Bai et al., 2022). For the former, the focus is largely on obtaining highly discriminative descriptors that are either optimised to be compact as in (Cao et al., 2020), or reduced to lower dimensionality with PCA as in (Hausler et al., 2021), a more complete overview is given in (Chen et al., 2023). A key difference with these works is that they rely on Euclidean embeddings, whereas hyperbolic embeddings naturally offer a more compressed representation, which comes in handy when few dimensions are desired.

For using fewer bits per embedding dimension, classical solutions are given by quantization techniques (Jacob et al., 2018), where representations are converted from float to discrete values. Traditionally, methods such as Product Quantization (Jégou et al., 2011) have been used for this purpose, but binarized networks (Lin et al., 2020; Zhu et al., 2020) have also gained attention in recent years. In particular, the recent popularity of large-scale models (Radford et al., 2021a) has led to an increasing demand for quantized networks, resulting in a series of quantization models specifically designed for large models (Liu et al., 2021; Yao et al., 2022). Inspired by these developments, we seek to bring these advantages of binarization to hyperbolic embeddings.

Another direction for compressing embeddings is given by hashing. Hashing focuses on efficiently encoding high-dimensional data into compact binary codes (Shen et al., 2018; Hoe et al., 2021; Shen et al., 2020). The goal for hashing is to learn a compact binary code that preserves the semantic similarity or structure of the original data, thus reducing the storage and computational costs for retrieval methods whilst preserving performance as much as possible. As this is a learning problem, the majority of emphasis is placed on finding the right optimization targets (Hoe et al., 2021). Unlike our proposed binarization approach, hashing requires optimization of binary codes for a specific set of embeddings, thereby limiting its flexibility. In contrast, we only focus on methods which can be efficiently applied to embeddings without further optimization.

**Hyperbolic representation learning.** Hyperbolic deep learning is centered around gradient-based optimization in hyperbolic space, which differs from the Euclidean operators commonly used in deep learning layers. Early success was obtained by successfully embedding the nodes of hierarchies as hyperbolic vectors, outperforming Euclidean embeddings. Nickel & Kiela (2017) introduce Poincaré Embeddings, where hierarchical nodes are positioned by pulling and pushing nodes based on parent-child relations. Ganea et al. (2018b) extend this idea through hyperbolic entailment cones, where child nodes should strictly fall under the cone spanned by parent nodes. Other hyperbolic embeddings include Sala et al. (2018) and Balazevic et al. (2019), who further explore these ideas for incomplete information and graphs, respectively.

To make the step towards deep learning in hyperbolic space, Ganea et al. (2018a) and Shimizu et al. (2021) introduce hyperbolic linear, recurrent, convolutional, and self-attention layers in the most commonly used model of hyperbolic space: the Poincaré ball model. These works have served as foundation for hyperbolic deep learning on graphs (Pan & Wang, 2021), dimensionality reduction (Chami et al., 2021), complex networks (Muscoloni et al., 2017), social media (Sawhney et al., 2021), etc. For more details on hyperbolic layers, we refer to the survey of Peng et al. (2021).

Hyperbolic learning has also been investigated in the image and video domain, as outlined by Mettes et al. (2023). Broadly in the visual domain, hyperbolic geometry has been shown to aid in a variety

of tasks, including image segmentation (Atigh et al., 2022), object detection (Lang et al., 2022), and video action recognition (Long et al., 2020). These works have in common that they leverage hierarchical knowledge to maximize the benefits from the hyperbolic space, embedding related concepts closer together, thereby allowing for more compact and powerful representations. These advances in hyperbolic learning, especially when it comes to computer vision, have shown that hyperbolic space is a strong, albeit computationally slow, alternative for learning representations. By binarizing hyperbolic embeddings, we can now also achieve fast search.

**Relationship to other hyperbolic models**. There are five isometric models for hyperbolic space (Cannon et al., 1997). In this paper, we focus on the Poincaré disk model as we find that it is highly suited for binarization as coordinates on the Poincaré ball are finite and each axis is symmetric, which allows us to use a simple binarization strategy.

## 3 BINARY HYPERBOLIC EMBEDDING

As part of our contributions, we first prove that with codebook $\mathbf{U}$, which encodes the representation $\mathbf{x}$ into binary format $\mathbf{x}^b$ such that $\mathbf{x} = \mathbf{U}\mathbf{x}^b$, we are able to construct the approximate equivalence between hyperbolic distance $d_{\mathbb{D}}(\cdot, \cdot)$ and hamming distance $d_{\mathbb{H}}(\cdot, \cdot)$. Then, using the Poincaré model for simplicity, we show how to binarize distances in hyperbolic space.

### 3.1 HYPERBOLIC BINARY EQUIVALENCE

To lay the groundwork for binarization in hyperbolic space we prove the approximate metric equivalence between hyperbolic and euclidean space, and show that we can obtain nearest neighbour equivalence between hamming and approximated hyperbolic distance. Let $\mathbf{x}, \mathbf{y} \in \mathbb{D}^d$ denote a pair of vectors in hyperbolic space, where $d_{\mathbb{D}}(\cdot, \cdot)$ measures the hyperbolic distance, and let $\mathbf{x}^b, \mathbf{y}^b \in \{0, 1\}^{nd}$ denote the binary code of $\mathbf{x}$ and $\mathbf{y}$ by quantizing each dimension of $\mathbf{x}$ and $\mathbf{y}$ into $n$ bits such that with codebook matrix $\mathbf{U} \in \mathbb{R}^{d \times nd}$, we have $\mathbf{x} \approx \mathbf{U}\mathbf{x}^b$. Given these vectors, we can prove the following:

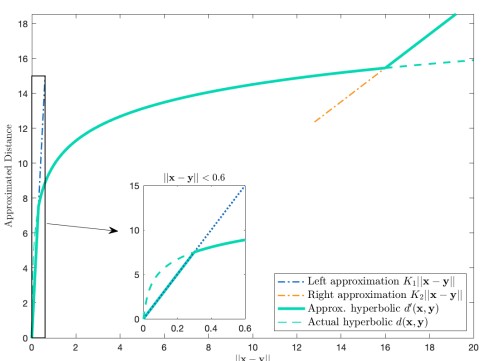

Figure 1: Approximated hyperbolic distance $d_{\mathbb{D}}'(\cdot, \cdot)$ on Poincaré ball. $d_{\mathbb{D}}'(\cdot, \cdot)$ is monotonically increasing to $\|\mathbf{x} - \mathbf{y}\|$

**Proposition 1** *Hyperbolic distance $d_{\mathbb{D}}(\mathbf{x}, \mathbf{y})$ is metric equivalent to Euclidean distance $d_{\mathbb{R}}(\mathbf{x}, \mathbf{y})$ by linear approximation.*

**Proof:** Hyperbolic distance in the unit Poincaré ball is defined as:

$$d_{\mathbb{D}}(\mathbf{x}, \mathbf{y}) = \cosh^{-1}\left(1 + 2\frac{\|\mathbf{x} - \mathbf{y}\|^2}{(1 - \|\mathbf{x}\|^2)(1 - \|\mathbf{y}\|^2)}\right). \tag{1}$$

Let $r_{\min} \leq \|\mathbf{x}\| \leq r_{\max}, r_{\min} \leq \|y\| \leq r_{\max}$, we have :

$$\cosh^{-1}\left(1 + 2\frac{\|\mathbf{x} - \mathbf{y}\|^2}{(1 - r_{\min}^2)^2}\right) \leq d_{\mathbb{D}}(\mathbf{x}, \mathbf{y}) \leq \cosh^{-1}\left(1 + 2\frac{\|\mathbf{x} - \mathbf{y}\|^2}{(1 - r_{\max}^2)^2}\right). \tag{2}$$

We can define the approximation of hyperbolic distance as:

$$d_{\mathbb{D}}'(\mathbf{x}, \mathbf{y}) = \begin{cases} K_1 \|\mathbf{x} - \mathbf{y}\|_2, & \text{if } \|\mathbf{x} - \mathbf{y}\|_2 \leq d_1, \\ \cosh^{-1}\left(1 + 2\frac{\|\mathbf{x} - \mathbf{y}\|^2}{(1 - \|\mathbf{x}\|^2)(1 - \|\mathbf{y}\|^2)}\right), & \text{if } d_1 < \|\mathbf{x} - \mathbf{y}\|_2 < d_2, \\ K_2 \|\mathbf{x} - \mathbf{y}\|_2, & \text{if } \|\mathbf{x} - \mathbf{y}\|_2 \geq d_2, \end{cases} \tag{3}$$

where at $\|\mathbf{x} - \mathbf{y}\|_2 \leq d_1$ and $\|\mathbf{x} - \mathbf{y}\|_2 \geq d_2$, the distance is approximated by a linear function of $\|\mathbf{x} - \mathbf{y}\|_2$. We set $d_1$ empirically and set $d_2$ to be the diameter of the Poincare ball so that this

approximation will never be triggered in practice. The relationship between $\|\mathbf{x} - \mathbf{y}\|$ and $d'_{\mathbb{D}}(\mathbf{x}, \mathbf{y})$ is illustrated in Figure 1.

The approximation can be made monotonically increasing by setting the slope of the linear approximation to:

$$K_1 = \cosh^{-1}\left(1 + \frac{2d_1^2}{(1 - r_{\min}^2)^2}\right) \Big/ d_1, \qquad K_2 = \cosh^{-1}\left(1 + \frac{2d_2^2}{(1 - r_{\max}^2)^2}\right) \Big/ d_2. \tag{4}$$

From the approximation, it follows that:

$$K_2 d_{\mathbb{R}}(\mathbf{x}, \mathbf{y}) \leq d'_{\mathbb{D}}(\mathbf{x}, \mathbf{y}) \leq K_1 d_{\mathbb{R}}(\mathbf{x}, \mathbf{y}), \tag{5}$$

where $d_{\mathbb{R}}(\mathbf{x}, \mathbf{y}) = \|\mathbf{x} - \mathbf{y}\|$ is the euclidean metric. Hence, under the approximation, the hyperbolic distance is metric equivalent to the Euclidean distance. $\qquad\square$

**Proposition 2** *For a codebook* $\mathbf{U}$ *such that* $\langle \mathbf{U}\mathbf{x}^b, \mathbf{U}\mathbf{y}^b\rangle \propto \langle \mathbf{x}^b, \mathbf{y}^b\rangle$, $d'_{\mathbb{D}}(\mathbf{x}, \mathbf{y})$ *is equivalent to hamming distance* $d_{\mathbb{H}}(\mathbf{x}^b, \mathbf{y}^b) = nd - \|\mathbf{x}^b \oplus \mathbf{y}^b\|_0$ *for nearest neighbor search.*

For any binary representation $\mathbf{x} \approx \mathbf{U}\mathbf{x}^b, \mathbf{y} \approx \mathbf{U}\mathbf{y}^b$, we have squared euclidean distance $d_{\mathbb{R}}$ is proportional to hamming distance $d_{\mathbb{H}}$:

$$d_{\mathbb{R}}^2(\mathbf{x}, \mathbf{y}) = \|\mathbf{x} - \mathbf{y}\|^2 = \|\mathbf{x}\|^2 + \|\mathbf{y}\|^2 - 2\langle \mathbf{x}, \mathbf{y}\rangle \tag{6}$$

$$= \|\mathbf{U}\mathbf{x}^b\|^2 + \|\mathbf{U}\mathbf{y}^b\|^2 - 2\langle \mathbf{U}\mathbf{x}, \mathbf{U}\mathbf{y}\rangle \tag{7}$$

$$\propto \|\mathbf{x}^b\|^2 + \|\mathbf{y}^b\|^2 - 2\langle \mathbf{x}^b, \mathbf{y}^b\rangle \tag{8}$$

$$= \lambda d_{\mathbb{H}}(\mathbf{x}^b, \mathbf{y}^b) \tag{9}$$

Then we have nearest neighbour equivalence between hamming distance $d_{\mathbb{H}}(\cdot, \cdot)$ and approximated hyperbolic distance $d'_{\mathbb{D}}(\cdot, \cdot)$:

$$\arg\min_{\mathbf{v}} d_{\mathbb{H}}(\mathbf{q}, \mathbf{v}) = \arg\min_{\mathbf{v}} d_{\mathbb{R}}^2(\mathbf{q}, \mathbf{v}) = \arg\min_{\mathbf{v}} d_{\mathbb{R}}(\mathbf{q}, \mathbf{v}) = \arg\min_{\mathbf{v}} d'_{\mathbb{D}}(\mathbf{q}, \mathbf{v}) \tag{10}$$

Intuitively, the propositions states that under a specific approximation, hyperbolic distance-based search generates the same output as the Hamming distance-based search, which can be computed quickly through binary operations.

## 3.2 Binary Quantization

With Proposition 2, we are able to perform binary operation based search, while using hyperbolic embeddings. Thus, we first generate full-precision hyperbolic embeddings on the poincaré ball and then binarize them via quantization.

**Hyperbolic embedding**. Given a training set $\mathcal{D}_{\text{train}} = \{(x_i, y_i)\}_{i=1}^{N}$, optionally equipped with label hierarchy $\mathcal{H}$. If a hierarchy $\mathcal{H}$ is provided, using Hyperbolic Entailment Cones (Ganea et al., 2018b), we have class prototypes $\mathbf{P} = [\mathbf{p}_1, \mathbf{p}_2, \cdots, \mathbf{p}_{|\mathcal{H}|}]$. Otherwise, we can simply set the class prototypes as maximum separated prototypes (Kasarla et al., 2022).

Next, with $\phi(y)$ the hyperbolic embedding of label $y \in \mathcal{Y}$, we train a network $f_\theta(\cdot)$ that projects the data point $v$ onto hyperbolic space $\mathbf{x} = f_\theta(x)$, where $f_\theta(\cdot)$ is an arbitrary backbone network equipped with hyperbolic embedding layer. The likelihood of sample $(x, y)$ is then given as:

$$p(y = y'|v) = \frac{\exp\left(-d_{\mathbb{D}}(f_\theta(x), \mathbf{p}_{y'})\right)}{\sum_{y''}^{|\mathcal{H}|} \exp\left(-d_{\mathbb{D}}(f_\theta(x), \mathbf{p}_{y''})\right)}, \tag{11}$$

which is optimized through the negative log-likelihood loss akin to Long et al. (2020).

**Binary quantization**. After hyperbolic embedding $\mathbf{x} = f_\theta(x)$, we perform binary quantization to obtain the binary representation of $\mathbf{x}$, denoted as $\mathbf{x}^b = g(\mathbf{x})$. In this section, we show that by designing a quantization matrix $\mathbf{U}$ such that it block-wisely satisfy $\langle \mathbf{U}\mathbf{x}^b, \mathbf{U}\mathbf{y}^b\rangle \approx \lambda\langle \mathbf{x}^b, \mathbf{y}^b\rangle$, we can exploit Proposition 2 for a binary Hamming distance-based search with hyperbolic embeddings.

In the Poincaré ball model, all dimensions fall in the radius of the ball $(-r, r)$. We shift each dimension by $r$ to make it in the range $(0, 2r)$:

$$\mathbf{x}^+ = \mathbf{x} + r. \tag{12}$$

This shift simplifies the calculations without changing the Euclidean distance:

$$d_{\mathbb{R}}(\mathbf{x}^+, \mathbf{y}^+) = \|\mathbf{x}^+ - \mathbf{y}^+\| = \|\mathbf{x} + r\mathbf{1} - (\mathbf{y} + r\mathbf{1})\| = \|\mathbf{x} - \mathbf{y}\| = d_{\mathbb{R}}(\mathbf{x}, \mathbf{y}). \tag{13}$$

In our proposed approach, the representation undergoes a dimension-wise quantization process. For n bits used by each dimension, we partition each dimension into a distinct set of $2^n - 1$ quantization levels under the same framework as (Jeon et al., 2020) with respect to the scale.

$$s = \frac{\sup(\mathbf{x}) - \inf(\mathbf{x})}{2^n - 1} = \frac{\sup(\mathbf{x}^+) - 0}{2^n - 1} = \frac{2r}{2^n - 1}, \tag{14}$$

where $\sup(\cdot)$ is the supremum and $\inf(\cdot)$ is the infimum. Then we can convert each dimension with respect to the scale into integers:

$$\mathbf{x}_{\text{int}} = \lfloor \frac{\mathbf{x}^+}{s} \rfloor, \tag{15}$$

which can be converted into *n-bits* binary code:

$$\mathbf{x}_{\text{int}} = \sum_{i=1}^{n} 2^{n-i} \cdot \mathbf{x}_i^b = 2^{n-1} \cdot \mathbf{x}_1^b + 2^{n-2}\mathbf{x}_2^b + \cdots + 2^0 \mathbf{x}_n^b, \tag{16}$$

where $\mathbf{x}_i^b \in \{0, 1\}^d$ represent the binary code for $i$-th significant bits in each dimension of $\mathbf{x}_{\text{int}}$. Subsequently, we can concatenate these bits to a binary representation, denoted as:

$$\mathbf{x}^b = \begin{pmatrix} \mathbf{x}_1^b \\ \mathbf{x}_2^b \\ \cdots \\ \mathbf{x}_n^b \end{pmatrix} \in \mathbb{R}^{nd}, \tag{17}$$

which results in:

$$\langle \mathbf{x}, \mathbf{y} \rangle \propto \langle \mathbf{x}_{\text{int}}, \mathbf{y}_{\text{int}} \rangle = 2^{n-1} \cdot \langle \mathbf{x}_1^b, \mathbf{y}_1^b \rangle + 2^{n-2} \cdot \langle \mathbf{x}_2^b, \mathbf{y}_2^b \rangle + 2^0 \cdot \langle \mathbf{x}_n^b, \mathbf{y}_n^b \rangle, \tag{18}$$

which is equivalent to $\langle \mathbf{U}\mathbf{x}_i^b, \mathbf{U}\mathbf{y}_i^b \rangle \propto \langle \mathbf{x}_i^b, \mathbf{y}_i^b \rangle$, leading to a block-wise application of Proposition 2. As such, we can apply the distance metric for binary hyperbolic embeddings as:

$$d_{\mathbb{R}}(\mathbf{x}, \mathbf{y}) \propto d_{\mathbb{H}}^{\Sigma}(\mathbf{x}^b, \mathbf{y}^b) = \Sigma_{i=1}^n 2^{n-i} \cdot d_{\mathbb{H}}(\mathbf{x}_i^b, \mathbf{y}_i^b) \tag{19}$$

$$= 2^{n-1} \cdot d_{\mathbb{H}}(\mathbf{x}_1^b, \mathbf{y}_1^b) + 2^{n-2} \cdot d_{\mathbb{H}}(\mathbf{x}_2^b, \mathbf{y}_2^b) + \ldots \cdot d_{\mathbb{H}}(\mathbf{x}_n^b, \mathbf{y}_n^b), \tag{20}$$

where $d_{\mathbb{H}}^{\Sigma}(\mathbf{x}^b, \mathbf{y}^b)$ is a summation of scaled hamming distance, hence we can use scaled binary hamming distance as an approximation of real valued distance. The scaling only happens on each of the $n - 1$ bits, resulting in $n - 1$ binary bit-shift operations with integer addition, which can be efficiently carried out. Equipped with a hyperbolic embedding network $f(\cdot)$ and binarization $g(\cdot)$, fast retrieval can be performed by embedding all elements in a collection with these functions. Then for a query $q$ and search collection $S$, both embedded to $\mathbb{D}$ and quantized, we can perform fast nearest neighbor search:

$$\arg\min_{\mathbf{v} \in S} d_{\mathbb{H}}^{\Sigma}(\mathbf{q}^b, \mathbf{v}^b) \stackrel{\triangle}{=} \arg\min_{\mathbf{v} \in S} d_{\mathbb{D}}(\mathbf{q}, \mathbf{v}). \tag{21}$$

## 4 EXPERIMENTS

### 4.1 EXPERIMENTAL SETUP

We focus on retrieval in both the image and video domain to measure the performance of various embedding compression approaches across multiple levels of compression (i.e., number of bits) in retrieval performance and speed. The performance is measured with mean average precision (mAP) and speed as the relative difference in retrieval time in seconds measured over the test set.

**Datasets.** For our experiments, we use three well-studied and hierarchical datasets: CIFAR100, ImageNet1K, and Moments in Time. CIFAR100 has 100 classes described by an officially defined hierarchy Krizhevsky et al. (2009), while for ImageNet1K each of the 1,000 object classes is a node in the WordNet hierarchy Fellbaum (1998). Similarly, for the Moments in Time dataset, each of

the 339 classes is a node in the VerbNet hierarchy Schuler (2006). CIFAR100 and ImageNet1K are image datasets, whereas Moments in Time is a video dataset.

**Implementation details.** For hyperbolic embedding learning, we use a curvature of $c = 0.1$ and the Riemann Adam optimizer Becigneul & Ganea (2019), supported by the *geoopt* Kochurov et al. (2020) library with a learning rate of $10^{-4}$. In practice, Riemannian Adam can be replaced by Adam when using the Poincaré disk model, as the learnable parameters of the backbones are in Euclidean space. All experiments were performed on a single Nvidia A6000 GPU. For the image experiments, we use the ResNet He et al. (2016) for CIFAR-100 and CLIP model with a ViT backbone on $32 \times 32$ patches (Radford et al., 2021a) for ImageNet1K. For the video experiments, we use the pre-trained 3D-ResNet backbone ResNeXtC3D Hara et al. (2018).

For fair comparison to baselines, the same frozen backbone is used across all competing models. Unless stated otherwise, we use two bits per dimension for binarization, following Hubara et al. (2018). Using one bit per dimension empirically did not achieve satisfactory results.

To measure the speed-up differences between binarized and non-binarized embeddings at different bit lengths, we perform stand-alone experiments solely for measuring the retrieval speed. Speed-up heavily relies on the implementation in the mathematical library used. For example, a boolean variable in Pytorch and Numpy is treated as an 8-bit unsigned int, which does not accurately reflect the speed-up. Therefore, we use a C++ implementation which supports both vectorized float operations and vectorized bitwise operations to evaluate the speed-up. All speed-ups are reported relative to the 512-dimensional full-precision representation.

### 4.2 BINARY VERSUS NON-BINARY RETRIEVAL

As a first experiment, we investigate the effect of manifold and binarization on retrieval performance and speed on all three datasets. For binarized embeddings, we use 512 bits and report the results for mAP@10. As our proof in Equation 21 shows that our binarization-based similarity is equivalent to the similarity in $\mathbb{R}$ and $\mathbb{D}$, we can use the same binarization strategy across all three manifolds: Euclidean $\mathbb{R}$, hyperspherical $\mathbb{S}$, and hyperbolic $\mathbb{D}$. For all manifolds, we use the same frozen backbone with a linear projection on top, supervised by the retrieval task, to get features with a desired dimensionality. The Euclidean baseline follows conventional cross-entropy optimization, while the hyperspherical baseline uses maximally separate prototypes Kasarla et al. (2022) optimized by reducing the pairwise cosine similarity between all prototype pairs. The hyperbolic embedding is trained without binarization, in correspondence with the prototypical hyperbolic learning approach of Long et al. (2020).

Table 1: **Comparing manifolds and binarization for retrieval** on CIFAR100, ImageNet1K, and Moments-in-Time. Underlined scores denote best full-precision embedding performance, **bold** scores denote best binary embedding performance. With full precision, hyperbolic embeddings already outperform Euclidean embeddings but are slow to evaluate. Our binary hyperbolic embeddings at 512 bits are able to maintain this performance while being much faster to evaluate, thereby getting the best of both worlds.

| Manifold | Binarized | CIFAR100 mAP@10 | ImageNet1K mAP@10 | Moments-in-Time mAP@50 | Speed |
|---|---|---|---|---|---|
| $\mathbb{R}^n$ Radford et al. (2021b) | ✗ | 0.674 | 0.576 | 0.140 | 1.00× |
| $\mathbb{S}^n$ Kasarla et al. (2022) | ✗ | 0.704 | 0.590 | 0.142 | 0.83× |
| $\mathbb{D}^n$ Long et al. (2020) | ✗ | 0.708 | 0.613 | 0.161 | 0.21× |
| $\mathbb{R}^n$ | ✓ | 0.665 | 0.556 | 0.119 | 4.71× |
| $\mathbb{S}^n$ | ✓ | 0.695 | 0.589 | 0.142 | 4.71× |
| $\mathbb{D}^n$ (ours) | ✓ | **0.706** | **0.608** | **0.158** | 4.71× |

The results in Table 1 show that for full-precision embeddings, hyperbolic space shows great promise when it comes to retrieval, outperforming its Euclidean and hyperspherical alternatives. However, hyperbolic embedding retrieval is five times slower compared to Euclidean retrieval. With binary hyperbolic embeddings, we are able to induce a large speed-up, at the same level of Euclidean and hyperspherical binarization, while obtaining the highest retrieval scores. On ImageNet1K, we

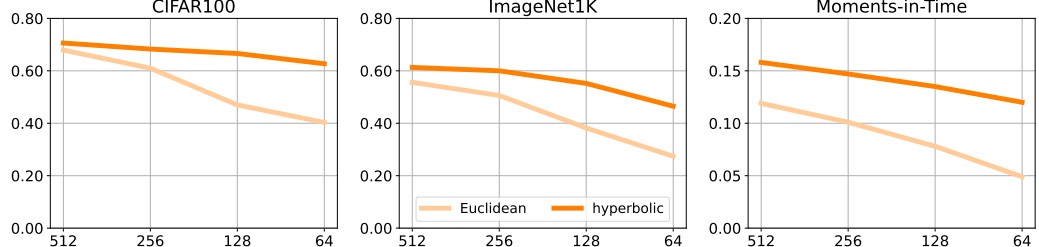

Figure 2: **Retrieval performance (mAP@10) as a function of the number of bits** on CIFAR100, ImageNet1K, and Moments-in-Time. Across the datasets, we find that hyperbolic embeddings allow for strong compression while maintaining performance, while Euclidean embeddings require more bits to maintain retrieval performance.

obtain an mAP@10 of 60.8% compared to 55.6% and 58.9% for the Euclidean and hyperspherical baselines. Across datasets, we find that hyperbolic embeddings retain more relative and absolute performance when binarized, highlighting the strong match between binary encodings and hyperbolic space. We conclude that binarization on top of hyperbolic embeddings is preferred for retrieval.

### 4.3 EFFECT OF BIT LENGTH AND QUANTIZATION LEVEL

**Effect of bit length.** A recurring theme in recent hyperbolic learning papers is the potential of low-dimensional effectiveness (Long et al., 2020; Ermolov et al., 2022; Ghadimi Atigh et al., 2021). To explore the strong impact of low-dimensional embeddings for fast retrieval, we have performed an experiment where we compare Euclidean to hyperbolic embeddings as a function of the number of used bits on all three datasets. For both baselines, we investigate using 512, 256, 128, and 64 bits, which corresponds to using 256-, 128-, 64-, and 32-dimensional embedding dimensions.

We show the bit length comparison in Figure 2. When using 512 bits, both embeddings maintain nearly all their performance compared to full-precision. When using fewer bits however, Euclidean embeddings drop in performance. Hyperbolic embeddings on the other hand are much less affected by the reduction in bits. At 64 bit embeddings, we obtain an mAP@10 of 62.7% on CIFAR100, a small drop compared to 70.6% at 512 bits. On the other hand, 64-bit Euclidean embeddings obtain an mAP@10 of 40.3%. This result indicates that with strong quantization and fewer bits, hyperbolic space is favored over the default Euclidean space.

**Effect of quantization level.** By quantizing each dimension into levels we can choose the bits per dimension. The total number of bits can therefore be determined by using more embedding dimensions with few bits or vice versa. For example, a 128-bit embedding can be obtained from a 64-dimensional embedding with 2 bits per dimension, or a 32-dimensional embedding with 4 bits per dimension. In Table 2, we show the impact across multiple choices of

Table 2: **The effect of embedding dimensions and quantization bits** on ImageNet1K. Underlined scores denote full-precision. We find it is best to use more dimensions with strong compression. With binary hyperbolic embeddings, we can obtain $> 8$ times faster at roughly the same performance. Our approach allows for larger speed-ups at the cost of retrieval performance.

| Bits $n \times d$ | mAP | Speed |
|---|---|---|
| 8 $\times$2 = 16 | 0.167 | 64.03$\times$ |
| 8 $\times$4 = 32 | 0.242 | 62.03$\times$ |
| 16 $\times$2 = 32 | 0.357 | 62.03$\times$ |
| 16 $\times$4 = 64 | 0.405 | 61.08$\times$ |
| 32 $\times$2 = 64 | 0.462 | 61.08$\times$ |
| 32 $\times$4 = 128 | 0.547 | 30.54$\times$ |
| 64 $\times$2 = 128 | 0.559 | 30.54$\times$ |
| 64 $\times$4 = 256 | 0.607 | 15.88$\times$ |
| 128$\times$2 = 256 | 0.608 | 8.25$\times$ |
| 128$\times$4 = 512 | 0.608 | 8.25$\times$ |
| 512$\times$32 = 16384 | 0.613 | 1.00$\times$ |

bits sizes. Overall, we find that it is beneficial to use more embedding dimensions with stronger compression than the other way around. This indicates that there is more information in any additional dimension compared to the additional precision within a dimension.

**How many bits do we need compared to full-precision Euclidean embeddings?** The ablations above, result in a natural question: how much can we speed up retrieval with binary hyperbolic embeddings with good performance? The speed-up results in Table 2 paint a clear picture: the first 8-fold speed-up can be obtained without hampering retrieval performance. Our approach allows for much bigger speed-up, but high compression then comes at the price of lower retrieval performance, making it a design choice how to balance both.

## 4.4 EFFECT OF HIERARCHICAL KNOWLEDGE

A key benefit of hyperbolic space is the potential to embed hierarchical knowledge with minimal distortion at much greater performance than Euclidean hierarchical embeddings Ganea et al. (2018b); Sala et al. (2018). This potential is reflected in the ability of a hyperbolic network to retrieve semantically similar items of adjacent classes. To measure this potential we follow Long et al. (2020); Ghadimi Atigh et al. (2021) in using the *Sibling mAP* (SmAP) performance metric. Building upon the mAP metric, SmAP takes into account the proximity in the class hierarchy for retrieved items. Specifically, when an item retrieved is just one hop away (i.e., same parent class) from the ground truth it is considered a true positive.

Table 3: **The effect of using hierarchical knowledge and different manifolds** on CIFAR100. At full-precision Spherical and hyperbolic embeddings outperform Euclidean embeddings, at the cost of a large number of bits and/or slow distance calculations. By binarizing we can maintain the performance benefits of hierarchy and hyperbolic embeddings but at a highly compressed bit length. $\star$ denotes that the method has been modified to be using hyperbolic distance metric.

| | Hierarchical | Manifold | Bit length | Binary | mAP | SmAP |
|---|---|---|---|---|---|---|
| Radford et al. Radford et al. (2021b) | ✗ | $\mathbb{R}$ | 16,384 | ✗ | 0.674 | 0.806 |
| Kasarla et al. Kasarla et al. (2022) | ✗ | $\mathbb{S}$ | 3,168 | ✗ | **0.706** | 0.824 |
| Kasarla et al. Kasarla et al. (2022)$\star$ | ✗ | $\mathbb{D}$ | 3,168 | ✗ | 0.708 | 0.821 |
| Barz et al. Barz & Denzler (2020) | ✓ | $\mathbb{S}$ | 3,200 | ✗ | 0.697 | 0.836 |
| Long et al. Long et al. (2020) | ✓ | $\mathbb{D}$ | 1,600 | ✗ | 0.708 | 0.840 |
| Binary (ours) | ✓ | $\mathbb{D}$ | 512 | ✓ | 0.706 | 0.838 |

In Table 3 we perform a comparison to gain insight into the effect of using hierarchy and hyperbolic embeddings. We show that prior full-precision approaches perform comparable in terms of mAP when using a hyperbolic or Spherical manifold, and that adding hierarchy is especially beneficial to the SmAP performance. However, despite being more compressed than full-precision Euclidean manifolds these approaches still require a large number of bits and/or slow hyperbolic distance calculations. By binarizing, we are able to compress hierarchical hyperbolic embeddings to a small bit length whilst maintaining good performance on both standard and hierarchical metrics.

**Qualitative analysis.** In Figure 3 we compare a non-hierarchical spherical space with our hierarchical hyperbolic space. All classes are connected to classes with pairwise cosine similarity greater than $0.5$. To measure the similarity between classes we average the embeddings for all instances of a class, reducing it to pair-wise relationships. From this visualization we can see that we are better at organizing concepts hierarchically, which as a consequence means that inputs with hierarchically similar concepts are more likely to fall in the same quantization bin. This enables better hierarchical performance even at low bit length. We suspect this is because spherical embeddings are learned by forcing classes to be equally dissimilar, whereas in hyperbolic space we can enforce a margin between classes while keeping track of siblings due to its infinite boundary nature.

## 4.5 EFFECT OF CURVATURE

The curvature and radius of the Poincaré disk model are determined by $c$, where higher values shrink the embedding space. Emprically, we found that a different $c_1$ value can be used when constructing (optionally with the hierarchy $\mathcal{H}$) the class prototypes $\mathbf{P}$ than when optimizing $f(\cdot)$ to map the input samples to the class prototypes where we use $c_2$. Meanwhile, $c_2$ is the curvature when we actually use the prototypes to generate hyperbolic embeddings for images and videos, it can be regarded as adjusting the hyperbolic metric, resulting in a different distance calculation with

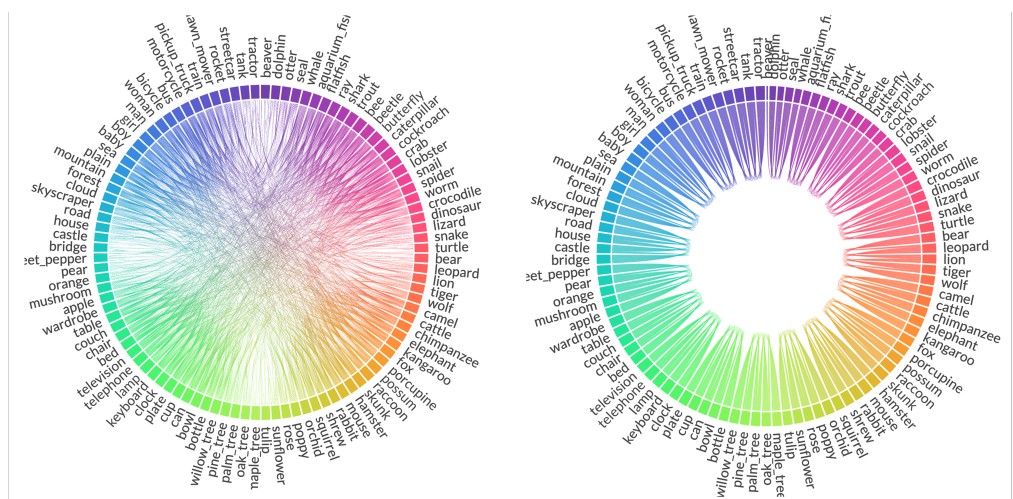

Figure 3: **Visualization of pair-wise class similarities** for (Left) non-hierarchical embeddings quantized from spherical space and (right) our hierarchical binary embeddings. Our embeddings are organized more hierarchically enabling stronger quantization with better hierarchical performance.

Table 4: **The effect of curvature** on CIFAR100. Parameter $c_1$ is the curvature used when constructing the hierarchy $\mathcal{H}$ to train class prototypes $\mathbf{P}$, whereas $c_2$ is the curvature used when optimizing $f(\cdot)$ to map data samples to class prototypes. Low curvature indicates almost uncurved, euclidean-like space, whereas high curvature causes numerical instability. Thus, a medium value of curvature during both class prototype embedding and sample embedding will be preferred.

|  | $c_1 = 0.001$ | $c_1 = 0.01$ | $c_1 = 0.1$ | $c_1 = 1$ | $c_1 = 10$ |
| --- | --- | --- | --- | --- | --- |
| $c_2 = 0.001$ | 0.613 | 0.635 | 0.695 | 0.702 | 0.694 |
| $c_2 = 0.01$ | undef | 0.691 | 0.666 | 0.705 | 0.698 |
| $c_2 = 0.1$ | undef | undef | 0.688 | **0.706** | 0.704 |
| $c_2 = 1$ | undef | undef | undef | 0.671 | 0.693 |
| $c_2 = 10$ | undef | undef | undef | undef | 0.507 |

the same prototypes. In Table 4 we compare different settings for $c_1$ and $c_2$ and find an interaction between the two parameters, but that for the settings compared, the performance is fairly stable, with the highest performance obtained with a high $c_1$ and a low $c_2$. Overall, it seems that training the class prototypes with a higher curvature is preferred, we suspect that this may be because the class prototypes are not pushed to the disk boundary, thereby leaving some room for embedding class instances in the later stage.

## 5 CONCLUSION

Hyperbolic deep learning has gained traction for a wide range of applications, from graphs to videos. However, its application in large-scale search has been hampered by slow distance calculations. In this work, we overcome this limitation by proving the equivalence between hyperbolic and Hamming distances, which allows us to binarize the hyperbolic space and significantly speed-up distance calculations. We experimentally verify this acceleration, across the video and image domain, obtaining at least $4.7\times$ faster at roughly equal performance. Our hyperbolic binary embeddings demonstrate the viability of hyperbolic space for large-scale retrieval and search, as well as opening the door to broader incorporation of hierarchical knowledge.

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
