# OpenReview forum: "Binary Hyperbolic Embeddings"
_ICLR.cc/2024/Conference — Submitted to ICLR 2024_

### Official Review · Reviewer_4fSp · 2023-10-22

**Soundness:** 3 good
**Presentation:** 3 good
**Contribution:** 3 good
**Rating:** 6
**Confidence:** 5

**Summary:**

This paper find a way, which builds a metric connection between hyperboic space and hamming space, to binary the hyperbolic embedding for fast retrieval with comparable mAPs as SOTA methods. Richful experiments validate the effectiveness of the proposed approach.

**Strengths:**

althrough hyperbolic embeddings can yield competitive retrieval performance (w.r.t. mAP@10) against other SOTA approaches, but with low computations for search. This paper proves that slow hyperbolic distance computation is equivalent to fast Hamming distance
computation, meanwhile maintain its good retrieval performance.

**Weaknesses:**

(1) Some of the derivation details, such as Eq.(6-9) in Proposition 2 and Eq. (11) with symbols not specified, are confusing and incorrect.
(2) Further explanations are needed on how to satisfy some preset conditions in proof or derivations.
(3) Refer to Questions.

**Questions:**

(1) What is the main difference between Hyperbolic embedding and learning to hashing?
(2) It's interesting to provide more experiments and analysis about the fastness and goodness of Hyperbolic embedding and learning to hashing.

---

> ### Author Response · Authors · 2023-11-23
>
> ### Reponse to Reviewer 4fSp
>
> We thank the reviewer for their positive comments on the performance and feedback on improving the understandability of the theory part.
>
> **In response to W1: Some of the derivation details, such as Eq.(6-9) in Proposition 2 and Eq. (11) with symbols not specified.**
>
> We thank the reviewer for the clarification suggestion, and we have added the notation explanations to the corresponding text.
>
> **In response to W2: Further explanations are needed on how to satisfy some preset conditions in proof or derivations.**
>
> For Proposition 1, the distance approximation preset condition requires constants $K_1$ and $K_2$, $K_2$ is usually set such that no point pairs would fall into that range. In practice, the model's performance is rather insensitive to $K_1$, even removing the approximation on $K_1$ side rarely harms the performance.
>
> For Proposition 2, the distance equivalence is based on a code book $\mathbf{U}$ satisfying $\langle\mathbf{Ux}^b, \mathbf{Uy}^b \rangle = \langle\mathbf{x}^b, \mathbf{y}^b \rangle$, in our case, our binarization blockwisely satisfied this, but in general any codebook, e.g. orthogonal codebook, can utilize Proposition 2.
>
> **In response to Q1: What is the main difference between Hyperbolic embedding and learning to hashing?**:
>
> In our case, we do not need to learn to achieve a performance that is comparable with full precision baseline, while hashing requires learning and solving a NP-hard discrete optimization problem, or its relaxed version at the cost of optimality.
>
> **In response to Q2: It's interesting to provide more experiments and analysis about the fastness and goodness of Hyperbolic embedding and learning to hashing.**
>
> For the same length of binary-code, learning to hash is slightly faster than us because hash-codes can be directly used for hamming distance, while in our case, extra bit shifts will be carried out for distance calculation. However, as shown in the following table, at equal bit length the accuracy of our approach is notably higher.
>
> |          | 16bits | 32bits | 64bits | 128bits | 256bits | 512bits |
> |----------|--------|--------|--------|---------|---------|---------|
> | OrthoCos [2] | 0.4937 | 0.5166 | 0.5608 | 0.5622  | 0.5577  | 0.5472  |
> | biHalf [3]   | 0.5242 | 0.6155 | 0.6957 | 0.7161  | **0.7396**  | 0.7347  |
> | Ours     | **0.6474** | **0.6769** | **0.7072** | **0.7198**  | 0.7297  | **0.7401**  |

---

### Official Review · Reviewer_AY1W · 2023-10-26

**Soundness:** 3 good
**Presentation:** 3 good
**Contribution:** 3 good
**Rating:** 6
**Confidence:** 3

**Summary:**

This paper proposes a hyperbolic space as a vector-based information retrieval. To keep the benefit of hyperbolic space while making the retrieval fast and reliable, the paper shows that hyperbolic distance computation can be replaced with the Hamming distance computation with proper binary encoding. Through the experiments with three datasets, the proposed method achieves less storage with a faster search while having competitive performance against Euclidean embeddings.

**Strengths:**

- The proposed method is well justified through the theoretical analysis. The experimental results support the theory.
- This paper is well-written and easy to follow.

**Weaknesses:**

- In experiments, it is written that prototypes and embeddings are learned with different curvatures. Different curvature means different hyperbolic space. Measuring the distance between two points in different spaces doesn’t make sense to me. Although I understand that having prototypes not located near the boundary would benefit the overall performance empirically, this approach cannot be justified in theory.
- I suspect that the paper is written in a hurry. Here are some editorial comments about the manuscript.
    - There are some capitalization errors here and there (e.g., euclidean, poincare)
    - Use proper latex command for citations (\citet and \citep).
    - Use the vector image for better quality (Figure 1)
    - Typo in equation (11) and above. I guess the argument for function f is v, not x.
    - It would be good to provide additional background on metric equivalence for a wider audience.

**Questions:**

- Is adaptive quantization not considered in this work? since the data points are likely to be located near boundaries, appropriate adaptive quantization may improve the performance.
- How many bits are needed to achieve the same performance as the full precision method (Table 2)?
- Is the comparison with the other methods fair? It is noted that C++ implementation is used for the proposed method. What about the other methods? Are they also implemented in C++? if not, can we say this is a fair comparison?
- How are the hyperbolic embeddings obtained? For example, how does the embedding of CIFAR-100 is obtained from ResNet? Are the Euclidean embeddings transformed via exponential mapping?

---

> ### Author Response · Authors · 2023-11-23
>
> We express our gratitude to the reviewer for their affirmative remarks on the theoretical soundness of our work and its congruence with experimental results.
>
> **In response to W1: Differences in Curvatures for Prototypes and Embeddings**:
>
> While our empirical data demonstrate effectiveness, we acknowledge the lack of rigor in our description. The performance degradation at the Poincaré ball's edge is attributed to numerical instability. Hence, we propose restricting the prototype's radius rather than using varying curvatures for embedding. This adjustment leads to the subsequent analysis, highlighting performance enhancement due to the adoption of a more robust CLIP backbone.
>
> |              | $r = \sqrt{10^3}$ | $c = \sqrt{10^2}$ | $c = \sqrt{10}$ | $r = 1$ | $r = \sqrt{0.1}$ |
> |--------------|-------------------|-------------------|-----------------|---------|-------------------|
> | $c_2 = 0.001$ | 0.653             | 0.676             | 0.735           | 0.742   | 0.734             |
> | $c_2 = 0.01$  | undefined         | 0.731             | 0.736           | 0.745   | 0.738             |
> | $c_2 = 0.1$   | undefined         | undefined         | 0.744           | **0.745** | 0.744             |
> | $c_2 = 1$     | undefined         | undefined         | undefined       | 0.700   | 0.733             |
> | $c_2 = 10$    | undefined         | undefined         | undefined       | undefined   | 0.527             |
>
> **In response to Q1: Potential of Adaptive Quantization in Enhancing Performance**:
>
> We fully agree with the comments, and we would be excited to see further research in this direction. We will include it in the conclusions.
>
> **In response to Q2: Bit Requirement for Parity with Full Precision Methods**:
>
> Generally, we observe that 256 bits suffice to achieve performance comparable to full precision methods. Refer to rows 9 and 11 in Table 2 for further details.
>
> **In response to Q3: Fairness of Comparison with Other Methods**:
>
> We ensure fairness in our comparative analysis, both in terms of software and hardware. We utilized the **same codebase** (C++ standard libraries) with **identical compilation options** and performed evaluations on the **same hardware**, specifically the Nvidia A6000 GPU for training and a AMD EPYC 7402P Processor (single-core, single-threaded only) for evaluation.
>
> **In response to Q4: Procedure for Obtaining Hyperbolic Embeddings**:
>
> The Euclidean embeddings are indeed transformed through exponential mapping, followed by a linear layer for dimensional adjustment. This method was applied, for example, to derive embeddings for CIFAR-100 from ResNet.

---

### Official Review · Reviewer_5iHW · 2023-10-31

**Soundness:** 3 good
**Presentation:** 3 good
**Contribution:** 3 good
**Rating:** 5
**Confidence:** 3

**Summary:**

- The authors prove the approximate equivalence between hyperbolic distance and Hamming distance, allowing fast binary operations for similarity search with hyperbolic embeddings.

- The authors propose a method to binarize vectors in the Poincaré ball model of hyperbolic space by quantizing each dimension and converting it to binary codes.

- The authors show experimentally that binary hyperbolic embeddings can achieve 4.7x speedup compared to full-precision Euclidean embeddings, while maintaining better retrieval performance.

- Across image and video datasets, the authors demonstrate that hyperbolic embeddings are much more robust to aggressive quantization/binarization compared to Euclidean embeddings.

**Strengths:**

The key contribution is enabling fast and compact binary codes for similarity search using hyperbolic embeddings, through an approximate equivalence to Hamming distance. This makes hyperbolic embeddings viable for large-scale retrieval applications.

**Weaknesses:**

- The proof of equivalence between hyperbolic and Hamming distance is approximate, and its accuracy depends on the linear approximation parameters. More analysis could be provided on the tightness of this approximation.

- The quantization and binarization scheme is simple and applied in a dimension-wise manner. More sophisticated methods like product quantization could potentially improve accuracy.

- Only the Poincaré ball model is evaluated. Extending the binary encoding ideas to other hyperbolic models like Lorentz could increase generality.

- The image and video datasets used are standard but small-scale.

- The ResNet and 3D ResNet backbones used are a bit dated.

- There is no comparison to other binary encoding methods like binary autoencoders or binary hashing.

- The speedup measurements use a C++ implementation. For fairer comparison, all methods should be benchmarked in the same codebase/hardware.

- The impact of factors like codebook design and learning hyperparameters could be investigated more thoroughly via ablation studies

**Questions:**

- The linear approximation of the hyperbolic distance has two parameters, K1 and K2. Could you provide some analysis on the tightness of this approximation and how it impacts the equivalence with Hamming distance?

---

> ### Author Response · Authors · 2023-11-23
> **Response to Reviewer 5iHW (Part 1)**
>
> We appreciate the reviewer's summary of the efficiency and compactness of our proposed method. We also thank the reviewer's suggestions for improving the paper, and we address them accordingly as follows:
>
> **In response to W1: More analysis on linear approximation**:
>
> We would love to show 1) the tightness of several $K_1, K_2$ and 2) despite the different tightness levels, the approximation does not change the rank of the retrieval output.
>
> 1) The linear approximation mainly contributes to theoretical completeness. In practice, using different $K_1$ and $K_2$ yields negligible difference compared with the none-approximated version. More specifically, $K_2$ has no effect on the distance calculation as none of the point pairs falls into that range. Varying $K_1$ does change the distance measure, while the effect is imperceptible, because both 1) the approximated distance and 2) the actual hyperbolic distance is very big compared to the euclidean counterparts.
>
> 3) The approximation is monotonically increasing with respect to hyperbolic distance $\|\mathbf{x}-\mathbf{y} \|$, that means, nearby samples in poincare ball is still nearby under approximated distance metric and far-away samples are still far away.
>
> **In response to W2: More sophisticated methods like product quantization**:
>
> We add the product quantization result for comparison:
>
> |                      | CIFAR100 |        |        | ImageNet1K |        |        |
> |:--------------------:|:--------:|:------:|--------|:----------:|:------:|--------|
> |                      |  128bit  | 256bit | 512bit |   128bit   | 256bit | 512bit |
> | Product Quantization |   0.521  |  0.537 | 0.550  |    0.357   |  0.415 | 0.457  |
> |         Ours         |   0.720  |  0.729 | 0.740  |    0.559   |  0.607 | 0.608  |
>
>
> **In response to W3: Other hyperbolic models like Lorentz**,
>
> we thank the review's consideration for a broader impact. The reason that we did not consider other hyperbolic models it that they do not satisfy that each dimension is 1) symmetical and 2) ranges is in $(-r, r)$.
>
> Although we do not binarize other hyperbolic models, they can still use our binarization via isometry defined in hyperbolic geometry [ref1], with a single coordinates transformation. In particular, we give one example on how Lorentz model transform to Poincar\'e model [ref2] in one line:
>
> * $(x_0,x_1,...,x_n)\mapsto (\frac{x_1}{1+x_0}, \frac{x_2}{1+x_0},...,\frac{x_n}{1+x_0})$
>
> Note that the above Lorentz $\mapsto$ Poincar\'e transformation is a isometry, meaning that the distance between Lorentz embeddings are equal to the distance between the resultant Poincar\'e embeddings. Therefore any Lorentz embedding based retrieval can benifit from our Poincar\'e binarization.
>
>
> **In response to W4: The image and video datasets used are standard but small-scale.**
>
> Following the reviewer's guidance, we have added the Quick Draw dataset for a large-scale experiment. In Quick Draw, we embed the raw 50 Million images with a simple MLP backbone plus Euclidean/Hyperbolic Head.
>
> The train/test split is split into a 1% training set and a 99% test set using the scikit-learn library with a random seed of 42. As the hierarchy information is not available for this dataset, we simply regard all the classes as the children of the "Root". The embedding dimensionality is 64, and we use 4 bits binarization. We perform retrieval on two randomly picked subsets (one small scale and one large scale) of the test set.
>
> |                        | QuickDraw-50K | QuickDraw-10M |
> |------------------------|------------|-----|
> | Euclidean Binarization | 0.2445     |   0.0407  |
> | Ours                   | 0.3149     |   0.0712  |
>
> The results above highlight that our approach generalizes to large scale settings. Thank you.

---

> ### Author Response · Authors · 2023-11-23
> **Response to Reviewer 5iHW (Part 2)**
>
> **In response to W5: The ResNet and 3D ResNet backbones used are a bit dated.**
>
> We have updated ResNet to CLIP and 3D ResNet to 3D-SWIN-Transfomer [ref3], leading to the updated results in the paper:
> The results show that with updated backbones, our binary hyperbolic embeddings outperform other geometries.
>
> **In response to W6: There is no comparison to other binary encoding methods like binary auto-encoders or binary hashing.**
>
> We obtained the following results when compared with hashing on CIFAR-100:
>
> |          | 64bits | 128bits | 256bits | 512bits |
> |----------|--------|---------|---------|---------|
> | OrthoCos[ref2] | 0.5650 | 0.5622  | 0.5577  | 0.5472  |
> | biHalf[ref3]   | 0.6957 | 0.7161  | **0.7396** | 0.7347  |
> | Ours     | **0.7072** | **0.7198**  | 0.7297  | **0.7401**  |
>
>
> Note that in our evaluation protocols, the query and database set are both used as the test set, while in standard hashing setting the database set is the entire training set. Across both datasets, for both 64 and 128 bit settings, we obtain noticeably higher performance. Moreover, hashing requires learning and solving a NP-hard discrete optimization problem, which we can avoid due to our binarization strategy.
>
> **In response to W7: For fairer comparison, all methods should be benchmarked in the same codebase/hardware**:
>
> To ensure fair comparison we benchmarked the performance using the same codebase, which is C++ std libraries, under the same compilation options, we also used the same hardware for all the evaluation, which is the Nvidia A6000 GPU. We will include this information in Section 4.1.
>
> **In response to W8-1: The impact of factors like codebook design could be investigated more thoroughly via ablation studies**:
>
> Our main contribution is to show that hyperbolic embeddings can be binarized to get the best out of the compactness of hyperbolic space and the speed of XOR operations. We consider codebook designs as fruitful future directions and we will include it in the conclusions.
>
> **In response to W8-2: learning hyperparameters could be investigated more thoroughly via ablation studies.**
>
> In the experiments, we examined three hyper-parameters, 1) dimensionality of embedding space, 2) the number of quantization bits, 3) curvature of hyperbolic space. To elimiate the ambiguity, in addition to those existing hyper-parameters, we also introduce the embedding radius limitation as another hyper-parameter instead of the original "curvature to generate prototypes". Note that increase of performance is due to using a stronger CLIP backbone.
>
> |              | $r = \sqrt{10^3}$ | $c = \sqrt{10^2}$ | $c = \sqrt{10}$ | $r = 1$ | $r = \sqrt{0.1}$ |
> |--------------|-------------------|-------------------|-----------------|---------|-------------------|
> | $c_2 = 0.001$ | 0.653             | 0.676             | 0.735           | 0.742   | 0.734             |
> | $c_2 = 0.01$  | undef             | 0.731             | 0.736           | 0.745   | 0.738             |
> | $c_2 = 0.1$   | undef             | undef             | 0.744           | **0.745** | 0.744             |
> | $c_2 = 1$     | undef             | undef             | undef           | 0.700   | 0.733             |
> | $c_2 = 10$    | undef             | undef             | undef           | undef   | 0.527             |
>
>
> [ref1] Cannon, James W., et al. "Hyperbolic geometry." Flavors of geometry 31.59-115 (1997): 2.
> [ref2] Radford, Alec, et al. "Learning transferable visual models from natural language supervision." ICML2021.
> [ref3] Liu, Ze, et al. "Video swin transformer." CVPR2022.

---

### Official Review · Reviewer_d3do · 2023-11-03

**Soundness:** 3 good
**Presentation:** 3 good
**Contribution:** 3 good
**Rating:** 6
**Confidence:** 3

**Summary:**

This paper investigates the topic of hyperbolic embeddings.
The authors introduce a novel technique for producing binary hyperbolic embeddings, aiming to reduce the storage and computational costs of conventional hyperbolic embeddings.
They show that the hyperbolic distance computation can be approximately equivalent to the scaled binary Hamming distance computation.
Through several experiments, the authors demonstrate the efficacy of the proposed approach in comparison to other embedding methods.

**Strengths:**

S1. **Innovative Approach:**
The idea of combining binary representations with hyperbolic geometry for embeddings presents a new avenue in the research area.

S2. **Efficiency:**
Binary embeddings can be more space-efficient and faster to compute, which is crucial for large-scale applications where memory and computational resources are limited.

S3. **Theoretical Insights:**
This work does not solely rely on empirical findings; it incorporates theoretical insights to provide foundational support for the proposed binary embedding.

**Weaknesses:**

W1. **Unclear How to Incorporate Hierarchical Knowledge:**
The experiments (e.g., Table 3) demonstrate the potential benefits of incorporating hierarchical knowledge, suggesting that the proposed embeddings can effectively leverage such information. However, the lack of a detailed explanation or illustration of how hierarchical knowledge is integrated into the embedding process could hinder the reproducibility and understanding of the method.

W2. **Clarity In Proposition 2:**
In reviewing the proof of Proposition 2, I was confused about the formula of the hamming distance, i.e.,
$d_{\mathbb{H}}(\boldsymbol{x}^b, \boldsymbol{y}^b) = nd - {\Vert \boldsymbol{x}^b \oplus \boldsymbol{y}^b \Vert}_0$.

Based on my understanding, it should be $d_{\mathbb{H}}(\boldsymbol{x}^b, \boldsymbol{y}^b) = {\Vert \boldsymbol{x}^b \oplus \boldsymbol{y}^b \Vert}_1$. Could you provide an illustration of this formula?

Moreover, I have identified a potential gap in the logical progression from Equation (8) to Equation (9). The transition between these equations is a critical step in the proof, and it appears that additional clarification or intermediate steps are needed to fully substantiate the authors' claims.

W3. **Generalizability Concerns**
I noticed that in Sections 4.3 to 4.5, the experimental results are presented using a single dataset. While the results are promising, they do not fully demonstrate the robustness and general applicability of the proposed method across diverse data scenarios.
It would be highly beneficial if the authors could expand their experimental evaluation to include all three datasets mentioned in the supplementary material. This would not only reinforce the validity of the claims made but also demonstrate the method's performance across different types of data and tasks.

**Questions:**

Regarding W1:

Q1: It would be beneficial if the authors could include a step-by-step illustration or a more detailed algorithmic description that explicitly shows how hierarchical information is processed and incorporated into the embeddings. Or, if there are any pre-processing steps or specific transformation techniques used to encode hierarchical knowledge into the binary embeddings, these should be clearly described.

Regrading W3:

Q2: Could the authors extend their experimental evaluation to include additional datasets as presented in the supplementary material?

In addition, I also found some typos when reviewing the paper. I enumerate some of them below:

In Page 4, Equation (7) should be $\approx {\Vert \boldsymbol{U} \boldsymbol{x}^b \Vert}^2 + {\Vert \boldsymbol{U} \boldsymbol{y}^b \Vert}^2 - 2{\langle \boldsymbol{U} \boldsymbol{x}^b, \boldsymbol{U} \boldsymbol{y}^b \rangle}$.

In Page 7, Table 2: Bits $n \times d \rightarrow$ Bits $d \times n$.

In Page 8, Table 3: The authors' names of the methods are duplicated.

**Details Of Ethics Concerns:**

No.

---

> ### Author Response · Authors · 2023-11-23
>
> We thank the reviewer for their positive feedback regarding the innovation, efficiency, and technical insights.
>
> **In response to W1/Q1**
>
> The hierarchical knowledge acts as supervision for the embedded learning. Specifically, we use the hierarchy over all classes to embed each class as a point in the Poincaré ball following [ref1]. Then, we optimize the image or video representations to be close to their class prototype in hyperbolic space. We further clarify this in Section 3.2.
>
> **In response to W2**
>
> We regret the confusion caused by symbol $\oplus$. We would like to show that your description of the hamming distance $d_{\mathbb{H}}(\mathbf{x}^b,\mathbf{y}^b)=\||\mathbf{x}^b \oplus \mathbf{y}^b\||_1$
>
> is equivalent to our description given as
>
> $d_{\mathbb{H}}(\mathbf{x}^b,\mathbf{y}^b)= nd -\||\mathbf{x}^b \oplus' \mathbf{y}^b\||_0$.
>
> For binary representations $\mathbf{x}^b, \mathbf{y}^b \in \{0, 1\}^{nd}$, it holds that $\||\mathbf{x}^b \oplus \mathbf{y}^b\||_1 = \||\mathbf{x}^b \oplus \mathbf{y}^b\||_0$. If we denote $\oplus$ as the element-wise XOR operation and $\oplus'$ as the element-wise XNOR operation, it holds that $\||\mathbf{x}^b \oplus \mathbf{y}^b\||_0=nd-\||\mathbf{x}^b \oplus' \mathbf{y}^b\||_0$. Combining these two steps, we have the equivalence $\||\mathbf{x}^b \oplus \mathbf{y}^b\||_1=nd-\||\mathbf{x}^b \oplus' \mathbf{y}^b\||_0$.
>
> Despite the equivalence, we agree that $\oplus$ is more commonly used as an XOR operator than XNOR; hence, we will follow the reviewer's suggestion and adapt the equation for more consistency with existing conventions.
>
> **In response to W3/Q2**
>
> In the revision, we have added the corresponding part of the results on different datasets to the appendix and will keep updating them. Those results are consistent across different datasets, proving the generalization capability of our model.
>
> To draw more interest regarding this question, inspired by reviewer **5iHW** , we show the capability of our proposed model on a large-scale dataset, "Quick, Draw!", which has **50 million sketch images** from 345 classes. We demonstrate the effectiveness of our binarization as follows:
>
> The train/test split is split into a 1% training set and a 99% test set using the scikit-learn library with a random seed of 42.
> As the hierarchy information is not available for this dataset, we simply regard all the classes as the children of the "Root".
> The embedding dimensionality is 64, and we use 4 bits binarization. We perform retrieval on two randomly picked subsets (one small scale and one large scale) of the test set.
>
> |                        | 50K database | 10M database |
> |------------------------|------------|-----|
> | Euclidean Binarization | 0.2445     |   0.0407  |
> | Ours                   | 0.3149     |   0.0712  |
>
>
> We also thank the reviewer for pointing out the typos; we have fixed them in the updated version.
>
> [ref1] Ganea, Octavian, Gary Bécigneul, and Thomas Hofmann. "Hyperbolic entailment cones for learning hierarchical embeddings." ICML, 2018.

---

### Author Response · Authors · 2023-11-23

We greatly appreciate the reviewers pointing out that our work "presents a new avenue in the research area", "incorporates theoretical insights to provide foundational support" (**d3do**), "enables fast and compact binary codes" (**5iHW**), "well justified through the theoretical analysis" (**AY1W**), and "... fast Hamming distance computation, while maintaining its good retrieval performance" (**4fSp**). In the following, we will address general points and expand on specific remarks made by the reviewers in individual responses.

### **Comparison to product quantization and learning to hash**

We compared product quantization [1] on CIFAR100 and ImageNet1K.

|                      | CIFAR100 |        |        | ImageNet1K |        |        |
|:--------------------:|:--------:|:------:|--------|:----------:|:------:|--------|
|                      |  128bit  | 256bit | 512bit |   128bit   | 256bit | 512bit |
| Product Quantization |   0.521  |  0.537 | 0.550  |    0.357   |  0.415 | 0.457  |
|         Ours         |   **0.720**  |  **0.734** | **0.740**  |    **0.559**   |  **0.607** | **0.608**  |

Both our method and the product quantization can be regarded as post-processing given existing feature vectors, while binary hyperbolic embeddings perform better than the product quantization.

We also conduct experiments to compare with recent learning to hash models:

|          | 16bits | 32bits | 64bits | 128bits | 256bits | 512bits |
|----------|--------|--------|--------|---------|---------|---------|
| OrthoCos [2] | 0.4937 | 0.5166 | 0.5608 | 0.5622  | 0.5577  | 0.5472  |
| biHalf [3]   | 0.5242 | 0.6155 | 0.6957 | 0.7161  | **0.7396**  | 0.7347  |
| Ours     | **0.6474** | **0.6769** | **0.7072** | **0.7198**  | 0.7297  | **0.7401**  |

Hyperbolic space's compact nature helps us to compete on par with Learning to hash's strength on low-bit performance and small-scale datasets. As learning to hash method is end-to-end instead of post-training on feature vectors, it is computationally expensive to extend to large-scale dataset, while in our case, we achieved comparable performance while using only feature-based post-training.

### **Fair compairison with c++ implementation**

To ensure fair comparison we benchmarked the performance using the
* **same codebase**, using C++ std libraries.
* **same compile options**; the compilation options are the same across all binary and real-valued evaluations.
* **same hardware**, for training we use a single Nvidia A6000 GPU, and for the evaluation we use an AMD EPYC 7402P Processor (single core single thread only).

Despite using the same set-up, we agree that C++ implementation might be less straight-forward for users. Thus, for the revision, we  implemented a PyTorch-based version for all comparisons. Making use of the highly parallelized nature of GPUs, we achieve an even more significant retrieval speed-up that can be as high as ~10X. We will include this additional analysis in the Appendix.

### **Gernalization to other dataset**

Following the reviewer's guidance, we have added the Quick Draw dataset for a large-scale experiment. In Quick Draw, we embed the raw 50 Million images with a simple MLP backbone plus Euclidean/Hyperbolic Head.

The train/test split is split into a 1% training set and a 99% test set using the scikit-learn library with a random seed of 42. As the hierarchy information is not available for this dataset, we simply regard all the classes as the children of the "Root". The embedding dimensionality is 64, and we use 4 bits binarization. We perform retrieval on two randomly picked subsets (one small scale and one large scale) of the test set.

|                        | QuickDraw-50K | QuickDraw-10M |
|------------------------|------------|-----|
| Euclidean Binarization | 0.2445     |   0.0407  |
| Ours                   | 0.3149     |   0.0712  |

The results above highlight that our approach generalizes to large scale settings. Thank you.

### **Typos, and insuffciently explained notations.**

We appreciate The reviewers feedback regarding the typos and notations in our theory that helps increasing the readability of our paper. We conducted a thorough proofreading to rectify these issues. We have revised the notations used in the propositions to be more explicit and self-explanatory.

---

### Meta-Review · Area_Chair_Xxc9 · 2023-12-12

**Metareview:**

This paper was borderline and none of the reviewers championed the paper, so unfortunately it did not make the cut in the end. Apart from some issues mentioned in the reviews (in particular regarding whether the comparison to other algorithms is fair), the paper is not particularly innovative, given the large amount of work on binary hashing, and more recently on hyperbolic embeddings.

**Justification For Why Not Higher Score:**

See metareview.

**Justification For Why Not Lower Score:**

N/A

---

### Decision · Program_Chairs · 2024-01-16

Reject